# Very Favorable vs. Favorable Risk Groups in Metastatic Renal Cell Carcinoma: A Step Toward Personalized Treatment

**DOI:** 10.3390/cancers17071076

**Published:** 2025-03-23

**Authors:** Yunus Emre Altıntaş, Oğuzcan Kınıkoğlu, Deniz Işık, Tuğçe Kübra Güneş, Uğur Özkerim, Tuğba Basoglu, Heves Sürmeli, Hatice Odabaş, Nedim Turan

**Affiliations:** 1Department of Medical Oncology, Kartal Dr. Lütfi Kırdar City Hospital, Istanbul 34865, Türkiye; ogokinikoglu@yahoo.com (O.K.); dnz.1984@yahoo.com (D.I.); ugur.ozkerim@hotmail.com (U.Ö.); basoglutugba@gmail.com (T.B.); hevessurmeli@hotmail.com (H.S.); odabashatice@yahoo.com (H.O.); turan.nedim@hotmail.com (N.T.); 2Department of Medical Oncology, Ümraniye Training and Research Hospital, Istanbul 34760, Türkiye; drtugcekubragunes@gmail.com

**Keywords:** metastatic renal cell carcinoma, IMDC risk classification, very favorable risk, tyrosine kinase inhibitors

## Abstract

Metastatic renal cell carcinoma is an aggressive malignancy that requires accurate risk stratification to guide treatment decisions and predict patient outcomes. Existing classification models categorize patients into favorable-, intermediate-, and poor-risk groups; however, they may not fully capture the heterogeneity within the favorable-risk category. This study aims to refine prognostic stratification by distinguishing a “very favorable” subgroup within the favorable-risk cohort. Through retrospective analysis, we demonstrate that patients classified as very favorable exhibit significantly longer survival and improved treatment responses compared to those in the favorable group. These findings suggest that a more nuanced risk classification could enhance clinical decision-making by optimizing therapy selection, minimizing overtreatment, and improving individualized patient management. Our study contributes to the evolving landscape of precision oncology and supports the need for further validation of refined prognostic models in metastatic renal cell carcinoma.

## 1. Introduction

Renal cell carcinoma (RCC) represents the most common malignancy of the kidney and accounts for approximately 90% of renal cancers, with clear-cell RCC being the predominant histological subtype [1]. Growing evidence indicates that physical activity, alcohol consumption, occupational exposure to trichloroethylene, and high parity in women may affect the risk of developing RCC. However, the extent to which each of these factors contributes to RCC incidence trends within a given population may vary depending on the prevalence of other risk and protective factors, and the level of awareness and effectiveness of managing predisposing conditions, as well as surveillance efforts and the incidental detection of preclinical tumors [2]. Despite advancements in early detection, a significant proportion of patients present with metastatic disease at initial diagnosis, necessitating systemic therapy. The prognosis of mRCC has historically been poor, with a five-year survival rate of only 8% for patients with distant metastases [3]. However, the introduction of targeted therapies, including tyrosine kinase inhibitors (TKIs) targeting the vascular endothelial growth factor (VEGF) pathway and immune checkpoint inhibitors (ICIs), has markedly improved patient outcomes [4].

Risk stratification plays a pivotal role in the management of mRCC, guiding treatment decisions and prognosis estimation. The IMDC model, originally proposed by Heng et al., has been widely validated and remains a cornerstone for prognostic classification in mRCC [5]. This model stratifies patients into favorable-, intermediate-, and poor-risk groups based on clinical and laboratory parameters, including hemoglobin levels, corrected calcium, Karnofsky performance status, neutrophilia, thrombocytosis, and time from diagnosis to systemic therapy initiation [6,7]. The prognostic significance of these factors was confirmed in large, multicenter studies, demonstrating that patients in the favorable-risk group had a median OS exceeding 43 months, whereas poor-risk patients had a median OS of only 7.8 months [5]. While the IMDC model has significantly contributed to risk assessment, recent efforts have focused on further refining favorable-risk patients into very favorable and favorable subgroups to better capture clinical heterogeneity and improve treatment selection. Given the differential responses to targeted therapy and immunotherapy observed across risk groups, refining these classifications may facilitate more precise therapeutic strategies in the era of VEGF–targeted agents and ICIs.

Recent efforts to refine risk stratification have led to the introduction of a new classification distinguishing between “very favorable” and “favorable” risk groups. This distinction aims to better capture heterogeneity within the traditionally defined favorable-risk category and potentially optimize treatment approaches. While favorable-risk patients have been shown to derive substantial benefit from VEGF–targeted therapy, the role of ICIs in this subset remains an area of active investigation [8]. Notably, the CheckMate 214 trial demonstrated that while nivolumab plus ipilimumab significantly improved survival in intermediate- and poor-risk patients, favorable-risk patients exhibited superior outcomes with sunitinib [9]. These findings suggest that more precise risk stratification is needed to individualize treatment selection.

In this study, we retrospectively analyze the clinical outcomes of patients classified under the newly defined very-favorable- and favorable-risk groups, aiming to elucidate differences in treatment response and survival. According to the reimbursement policies in our country, first-line treatment in clinical practice predominantly consists of TKIs as the use of ICIs is not routinely covered. Consequently, this study provides valuable real-world insights into treatment outcomes where VEGF–targeted therapy remains the standard first-line option. Furthermore, we evaluate the clinical relevance of very-favorable-risk classification and compare treatment outcomes between very-favorable- and favorable-risk groups. By leveraging contemporary risk stratification models, we seek to refine therapeutic decision-making in mRCC and contribute to the evolving landscape of precision oncology.

## 2. Materials and Methods

### 2.1. Patient Population and Data Collection

This retrospective study included patients diagnosed with mRCC at Kartal Dr. Lütfi Kirdar city hospital in Istanbul, Türkiye between February 2017 and September 2023. Tumor staging was performed according to the Tumor-Node-Metastasis (TNM) classification system, which remains the standard for staging renal cell carcinoma. The TNM system evaluates tumor size (T), lymph node involvement (N), and distant metastasis (M) to determine disease stage and prognosis. This classification aids in risk stratification and treatment planning for patients with mRCC [10]. Out of 189 patients, 75 patients who met the criteria for the favorable-risk group as defined by the IMDC were included in the analysis. Three parameters were used to further divide the IMDC favorable-risk group into two categories: very favorable (patients with none of the risk factors) and favorable (patients with at least one risk factor) [7,8]. These parameters included the following:Time from diagnosis to systemic therapy (<3 years vs. ≥3 years);Karnofsky performance status (KPS; ≤90 vs. >90);Presence of brain, liver, or bone metastasis.

Based on these criteria, 29 patients were categorized into the very favorable group and 46 patients into the favorable group. Patients with intermediate- or poor-risk profiles were excluded from the study.

Data of patients diagnosed with mRCC were collected from the Kartal Dr. Lütfi Kirdar City hospital database. The selection process was based on the following inclusion and exclusion criteria: Inclusion criteria: patients with mRCC who received treatment with TKIs and had sufficient data available for calculation of the IMDC risk score. Exclusion criteria: patients who were treated solely with cytotoxic chemotherapy, and patients with prior or concurrent malignancies other than mRCC were excluded to ensure a homogenous study population, and those under the age of 18 were also excluded. All laboratory values were adjusted according to institutional reference ranges. The IMDC risk score was determined based on laboratory and clinical parameters recorded at the initiation of TKI therapy.

Due to the reimbursement limitations in our country, all patients included in this study were treated with single-agent TKIs as a first-line therapy. The two most commonly administered TKIs were sunitinib and pazopanib. Data on clinical parameters, histology, treatment history, and metastatic sites were collected retrospectively from patient medical records.

Patients in this study received either sunitinib or pazopanib for the treatment of metastatic renal cell carcinoma. Sunitinib was administered at a dose of 50 mg once daily following a 2-weeks-on 1-week-off regimen within a 3-week treatment cycle, while pazopanib was given at a dose of 800 mg once daily. Treatment with both agents continued until disease progression or the occurrence of unacceptable toxicity.

### 2.2. Outcomes

Treatment responses were evaluated every three months until disease progression, death, or loss of follow-up for patients who discontinued for other reasons, following the Response Evaluation Criteria in Solid Tumors [11]. PFS was determined as the number of months from the initiation of first-line treatment to disease progression or death, whichever occurred first. OS was defined as the number of months from the start of first-line treatment until death.

### 2.3. Statistical Analysis

Survival data analyses were conducted with 95% confidence intervals (CIs) calculated using the exact method. Clinical and pathological characteristics of the patients were compared using chi-square or Fisher’s exact tests, depending on the distribution of the data. A *p*-value of less than 0.05 was considered statistically significant. Kaplan–Meier survival curves were generated to evaluate progression-free survival (PFS) and overall survival (OS), and a log-rank test was used to compare survival outcomes between groups.

To identify factors associated with PFS and OS, univariate and multivariate Cox proportional hazards regression models were employed. Variables with a *p*-value below 0.10 in the univariate analysis, as well as those deemed clinically significant, were included in the multivariate analysis. As this study was retrospective, the sample size was determined by the number of eligible patients within the study period, and no formal sample size calculation was performed. Patients with missing data were excluded from the survival analysis.

All statistical analyses were performed using SPSS Statistics version 26.0 (IBM Corporation, Armonk, NY, USA).

## 3. Results

### 3.1. Clinical and Demographic Characteristics of mRCC Patients

A total of 189 patients with mRCC were included in the analysis. According to IMDC criteria, 75 patients were categorized into the favorable-risk group and were subsequently included in the analysis. Based on the novel classification system, 46 patients (61.3%) were assigned to the favorable-risk group, while 29 patients (38.7%) were classified into the very-favorable-risk group.

The clinical characteristics and treatment outcomes of patients in the very favorable and favorable groups were compared to assess differences in disease presentation, treatment patterns, and survival outcomes. There were no significant differences between the groups in terms of age, sex, histological type, Fuhrman grade distribution, prior nephrectomy, or choice of systemic treatment (*p* > 0.05).

As expected, based on our classification criteria, all patients in the very favorable group had a time to systemic therapy of ≥3 years, while the majority of the favorable group had a time to therapy of <3 years (*p* < 0.01). This parameter was used as a stratification criterion rather than an independent finding. All patients in the very favorable group developed metastases ≥3 years after their initial diagnosis, whereas 82.6% of patients in the favorable group developed metastases within 3 years.

By definition, patients in the very favorable group had no liver, bone, or CNS metastases, while these metastases were observed only in the favorable group. This reflects the stratification criteria rather than a novel observation. Lung metastases were more frequently observed in the very favorable group (69.0%) compared to the favorable group (45.7%), but this difference did not reach statistical significance (*p* = 0.059). CNS metastases were rare and reported only in the favorable group (8.7%), though this was not statistically significant (*p* = 0.15). The baseline characteristics of the two groups are summarized in Table 1.

### 3.2. Survival Analyses

The median duration of follow-up was 33.5 months (range = 27.8–39.3). The entire group had a median PFS of 16.0 months (95% CI, 12.2–19.7) and a median OS of 54.0 months (95% CI, 39.0–68.9).

In univariate analysis, the very-favorable-risk group demonstrated better PFS and a significantly lower risk of progression than the favorable-risk group (22.8 vs. 13.8 months, HR: 0.55, 95% CI: 0.33–0.91, *p* = 0.020). However, in multivariate analysis, the favorable-risk group did not retain statistical significance as an independent prognostic factor for PFS (HR: 0.58, 95% CI: 0.31–1.05, *p* = 0.071) after adjusting for other confounding variables (Table 2). The Kaplan–Meier estimates of PFS are shown in Figure 1.

In univariate analysis, the very-favorable-risk group demonstrated better OS and a significantly lower risk of death than the favorable-risk group (74.4 vs. 42.7 months; HR: 0.38, 95% CI: 0.17–0.81, *p* = 0.013). Similarly, in multivariate analysis, the favorable-risk group remained an independent prognostic factor for OS (HR: 0.34, 95% CI: 0.14–0.80, *p* = 0.014) after adjusting for confounding variables (i.e., age, sex, histological type, and systemic treatment) (Table 3). The Kaplan–Meier estimates of OS are shown in Figure 2.

## 4. Discussion

The findings of this study provide further validation for the concept of a very-favorable-risk group in mRCC as previously suggested by Schmidt et al. [8]. Our results demonstrate that patients classified as very favorable have significantly better OS and PFS compared to those in the favorable group. This distinction suggests that the existing IMDC favorable-risk group encompasses patients with a wide spectrum of prognostic outcomes, warranting a more refined classification to better guide treatment decisions.

The treatment landscape for mRCC has rapidly evolved with the integration of ICIs and TKIs, significantly improving survival outcomes. While ICI-TKI combinations have demonstrated efficacy in both intermediate- and poor-risk patients, the outcomes in favorable-risk patients remain variable, suggesting that a uniform classification may not accurately capture prognostic differences within this group [4,9,12,13,14]. Our findings further support the need for a refined risk stratification model, distinguishing a very favorable subgroup within the favorable-risk category. Such a classification could guide more personalized treatment approaches, ensuring that patients receive the most appropriate therapeutic strategy based on their individual risk profile.

The findings of our study align with emerging evidence incorporating metastatic sites into prognostic models for mRCC. Massari et al. demonstrated that brain, bone, or liver metastases as the primary site of metastatic disease significantly impact OS and PFS, underscoring prognostic heterogeneity within IMDC risk categories [15]. Including metastatic sites as an independent variable improved prognostic stratification, suggesting that the biological aggressiveness of tumors with early metastases to these organs is associated with worse outcomes. In our cohort, patients in the favorable-risk group with bone, liver, or brain metastases had a significantly poorer prognosis. This suggests that IMDC criteria should be reevaluated. Including metastatic sites in risk classification could improve prediction accuracy, leading to more personalized treatment strategies and better clinical decision-making.

A study by Teishima et al. investigated the prognostic value of the systemic immune-inflammation index (SII) in metastatic renal cell carcinoma (mRCC) patients and contributed to the reclassification of the IMDC risk model [16]. Their findings highlight the role of inflammatory markers in prognosis and further underscore the heterogeneity of mRCC patients. While our study did not evaluate inflammatory markers, the fact that they can influence risk stratification demonstrates the complexity and heterogeneity within the favorable-risk group.

Another important factor that separates the very favorable group is their physical performance. These patients had a higher KPS, showing that physical function plays a key role in mRCC outcomes. Results from a multicenter study have shown that performance status is an important predictor of survival. Since a KPS ≥80 was a defining feature of the very favorable group, these patients may have better physical strength, helping them tolerate and respond to treatment more effectively [6]. In our study, the very favorable group also demonstrated significantly longer OS and PFS compared to the favorable group, further supporting the impact of functional performance on prognosis.

Our findings are similar to those of Yekedüz et al., who validated the updated IMDC risk model by dividing the favorable-risk group into very favorable and favorable subgroups. Their study reported a median OS of 55.8 months in the very favorable group and 34.2 months in the favorable group (*p* = 0.025), with a median PFS of 25.5 months and 15.5 months, respectively (*p* = 0.010) [17]. In our study, OS was 74.4 months in the very favorable group and 42.7 months in the favorable group (*p* = 0.010), while PFS was 22.8 months and 13.8 months, respectively (*p* = 0.018). Both studies confirm the survival advantage of the very favorable group, but our results show longer OS in both groups, which may be due to differences in treatment strategies, patient selection, or follow-up duration. These findings indicate the variation within the IMDC favorable-risk group and lead to better risk model stratification to guide a treatment decision.

Our study’s findings match with those of Schmidt et al., who reported a median OS of 64.8 months in the very favorable group and 45.6 months in the favorable group (*p* < 0.001). Similarly, median PFS was significantly longer in the very favorable group compared to the favorable group (*p* < 0.001) [18]. These results are parallel to our findings, further supporting the prognostic value of refining the IMDC favorable-risk classification. The consistency between studies underscores the need for more precise risk models to better guide treatment decisions in metastatic renal cell carcinoma.

Looking at some key studies, our findings show that the new risk classification of mRCC is consistent with other studies. CheckMate 214 showed the superiority of nivolumab-ipilimumab over sunitinib in intermediate- and poor-risk mRCC, but favorable-risk patients had better progression-free survival with sunitinib, highlighting the need for further subclassification [9]. Our study refines this by identifying a very favorable subgroup with significantly longer survival, reinforcing the heterogeneity within the IMDC favorable-risk group. As CheckMate 214 focused on intermediate- and poor-risk patients, our results suggest that very favorable patients may benefit from a more conservative approach with TKIs rather than early immunotherapy.

The final analysis of the JAVELIN Renal 101 trial confirmed that a combination of avelumab and axitinib demonstrated prolonged PFS and a higher objective response rate (ORR) compared to sunitinib. However, the median OS difference did not reach statistical significance, likely due to a high rate of subsequent ICI treatment in the sunitinib arm, which may have affected the OS benefit. A more detailed breakdown of JAVELIN Renal 101’s results revealed that in favorable-risk patients, median OS was numerically longer with avelumab plus axitinib (79.4 months) versus sunitinib (65.5 months), but this difference was not statistically significant (HR: 0.73, 95% CI: 0.48–1.10; *p* = 0.1290). In contrast, significant OS benefits were observed in the poor-risk group, where avelumab plus axitinib demonstrated a 21.3-month median OS compared to 11.0 months with sunitinib (HR: 0.58, *p* = 0.0076) [19]. These findings suggest that while combination therapy may provide significant advantages in poor-risk patients, its benefits in favorable-risk patients remain uncertain. Our study, which focuses on the distinction between very favorable and favorable groups, further underscores this heterogeneity. The observation that patients in the very favorable group achieve significantly better survival outcomes suggests that not all favorable-risk patients should be treated as a uniform category.

The Cattrini et al. study is a network meta-analysis that evaluates first-line treatment options for mRCC by comparing ICI-based combinations and TKI monotherapy across different IMDC risk groups [20]. While combination regimens have demonstrated superior efficacy over sunitinib in intermediate- and poor-risk groups, their benefit in favorable-risk patients is less clear, suggesting that a one-size-fits-all approach may not be appropriate. Our study provides a contribution to this evolving paradigm by identifying a very favorable subgroup within the favorable-risk category. There is a necessity for a more precise classification system to guide personalized therapy selection and avoid potential overtreatment in patients who may benefit from a less intensive treatment like only TKI-based therapy.

This study has several limitations that should be acknowledged. First, its retrospective design introduces inherent biases, including selection bias and missing data, which may influence the reliability of our findings. The retrospective nature also limits the ability to establish causal relationships between prognostic factors and outcomes.

Second, our study was conducted in a single-center setting, which may reduce its generalizability to broader patient populations. Differences in institutional treatment practices and patient management approaches may also influence survival outcomes, limiting external validity.

Third, treatment selection bias must be considered as all patients receive TKIs as a first-line therapy due to reimbursement restrictions in our country. Consequently, the potential impact of ICIs in patients with very-favorable- or favorable-risk profiles could not be evaluated. Given the growing role of combination therapies in mRCC, future studies incorporating ICIs are needed to validate our findings in different therapeutic settings.

Another important limitation is the relatively small sample size, particularly in the very-favorable-risk group. While our results demonstrate significant differences in survival between the favorable and very favorable groups, a larger cohort would enhance statistical power and provide more robust conclusions. Regarding familial accumulation and genetic factors, as this is a retrospective analysis, such data were not available in patient records. Therefore, we cannot draw conclusions about hereditary or genetic predispositions to kidney cancer in our cohort.

## 5. Conclusions

In conclusion, our study demonstrates that subclassifying favorable-risk mRCC patients into very favorable and favorable groups provides important prognostic insights. Patients in the very favorable group exhibited significantly longer median OS and median PFS compared to those in the favorable group, emphasizing the heterogeneity within the IMDC favorable-risk category. These findings support the need for refined risk stratification to guide treatment decisions more precisely.

While ICIs have shown significant survival benefits in intermediate- and poor-risk patients, their superiority over TKIs in favorable-risk patients remains uncertain. Recent trials and network meta-analyses suggest that some favorable-risk patients may achieve optimal outcomes with TKI monotherapy, avoiding potential overtreatment. Further prospective studies are needed to validate these findings and explore whether incorporating additional biomarkers could enhance risk stratification in mRCC.

## Figures and Tables

**Figure 1 cancers-17-01076-f001:**
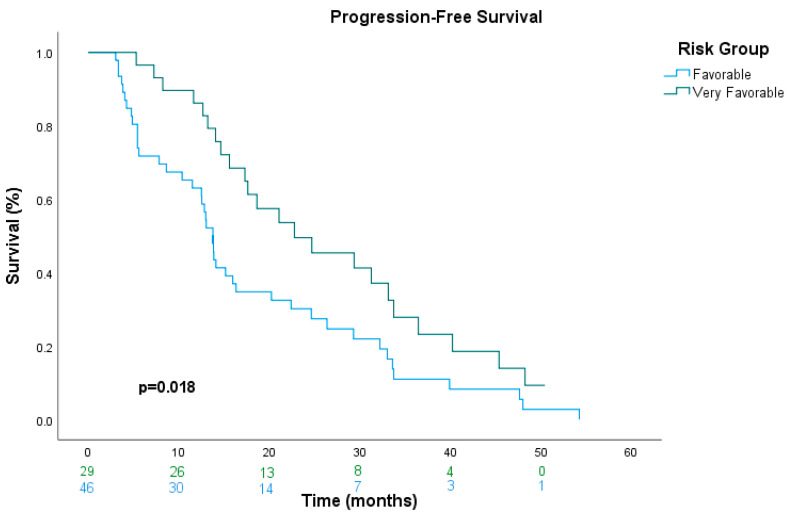
Kaplan–Meier curves indicating a significant improvement in progression-free survival in the very-favorable-risk group compared to the favorable-risk group.

**Figure 2 cancers-17-01076-f002:**
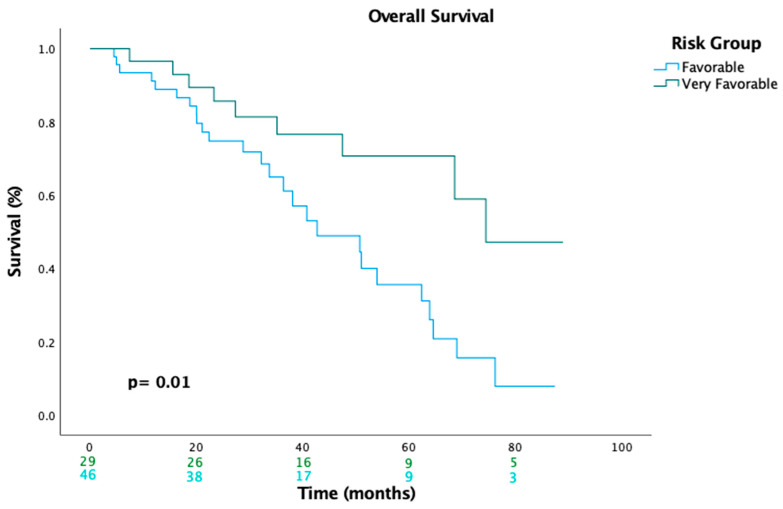
Kaplan–Meier curves indicating a significant improvement in OS in the very-favorable-risk group compared to the favorable-risk group.

**Table 1 cancers-17-01076-t001:** Clinical and demographic characteristics of metastatic renal cell carcinoma patients.

	Favorable		Very Favorable		*p*
	*n* = 46	(%)	*n* = 29	(%)	
Age, median (IQR) ^1^	62 (54–68)		60 (56–67)		0.19
Sex					0.99
Male	34	73.9	22	75.9	
Female	12	26.1	7	24.1	
Histological type					0.47
Clear cell	39	84.8	27	93.1	
Non-clear cell	7	15.2	2	6.9	
Fuhrman grade					0.88
1–2	13	28.3	8	27.6	
3–4	18	39.1	10	34.5	
Missing	15	32.6	11	37.9	
Previous nephrectomy					0.11
Yes	36	78.3	27	93.1	
No	10	21.7	2	6.9	
Systemic treatment					0.47
Sunitinib	27	58.7	20	69.0	
Pazopanib	19	41.3	9	31.0	
Metastatic sites					
Lung	21	45.7	20	69.0	0.059
Bone	11	24.0	0	0	<0.01
Liver	10	21.7	0	0	<0.01
Central nervous system	4	8.7	0	0	0.15
Time to systemic treatment					<0.01
<3 y	38	82.6	0	0	
≥3 y	8	17.4	29	100	

^1^ Abbreviation: IQR = interquartile range.

**Table 2 cancers-17-01076-t002:** Univariate and multivariate analysis for progression free survival in the favorable- and very-favorable-risk group.

	Univariate		*p* Value	Multivariate		*p* Value
	HR	95% CI		HR	95% CI	
Age			0.222			0.285
<65	1			1		
≥65	1.42	0.81–2.52		1.37	0.78–2.43	
Sex			0.567			0.806
Male	1.18	0.67–2.05		1.09	0.54–2.22	
Female	1			1		
Histological type			0.150			0.143
Clear cell	1			1		
Non-clear cell	0.59	0.29–1.21		0.545	0.24–1.23	
Systemic treatment			0.360			0.303
Sunitinib	1.27	0.76–2.10		1.372	0.75–2.50	
Pazopanib	1			1		
IMDC risk			0.020			0.071
Very favorable	1			1		
Favorable	0.55	0.33–0.91		0.58	0.31–1.05	

**Table 3 cancers-17-01076-t003:** Univariate and multivariate analysis for overall survival in the favorable- and very-favorable-risk group.

	Univariate		*p* Value	Multivariate		*p* Value
	HR	95% CI		HR	95% CI	
Age			0.198			0.221
<65	1	0.78–3.37		1		
≥65	1.62			1.60	0.75–3.40	
Sex			0.732			0.913
Male	1.15	0.51–2.60		0.94	0.35–2.57	
Female	1			1		
Histological type			0.850			0.723
Clear cell	1			1		
Non-clear cell	0.90	0.32–2.57		0.797	0.23–2.80	
Systemic treatment			0.895			0.890
Sunitinib	0.95	0.48–1.95		0.943	0.41–2.19	
Pazopanib	1			1		
IMDC risk						
Very favorable	1		0.013	1		0.014
Favorable	0.38	0.17–0.81		0.34	0.14–0.80	

## Data Availability

While the datasets analyzed in this study are not publicly accessible, they can be obtained from the corresponding author upon reasonable request.

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
