# Peer review of "Very Favorable vs. Favorable Risk Groups in Metastatic Renal Cell Carcinoma: A Step Toward Personalized Treatment"

_cancers, 2025, doi:10.3390/cancers17071076_

Round 1
Reviewer 1 Report
Comments and Suggestions for Authors
I read your manuscript with interest and agree that there may be a role for subdividing the current favorable risk category. It does appear that the criteria used to make the assignments of favorable vs very favorable have a prognostic significance.
However, the overall logic appeared to be circular in some places. For example, if you define very favorable risk as "time from diagnosis to systemic therapy as >3 years (line 105), then you cannot report in results, "a significant difference was observed in the time to systemic treatment..." Similarly, if you have defined very favorable as "absence of brain, liver or bone metastases," then it is questionable at best to report the significance of the fact that "liver and bone metastases were exclusively observed in the favorable group." (line 170) or CNS metastases were reported only int eh favorable group (line 174).These should be true and understood based on your own definition of very favorable risk.
Author Response
This document contains my responses to the reviewers' comments. To enhance clarity, each reviewer's comments have been highlighted in different colors:
Red: Reviewer 1 – Provided feedback on logical consistency, particularly regarding the classification criteria and how certain statistical findings were reported. I have addressed these concerns by clarifying that these differences were expected based on our predefined criteria and should not be interpreted as independent findings.
Blue: Reviewer 2 – Recommended adding the number of patients at risk in the Kaplan-Meier curves. I have acknowledged this valuable suggestion and incorporated the required information into the revised figures to improve the clarity and interpretability of the survival analysis.
Green: Reviewer 3 – Suggested expanding the discussion on risk factors for kidney tumors, ensuring references for the IMDC method, Karnofsky assessment, and TNM classification, and addressing missing clinicopathological data. I have incorporated additional details where necessary, ensured proper citations, and clarified the study methodology.
I have made every effort to address the reviewers' comments comprehensively and improve the manuscript accordingly. I appreciate their constructive feedback, which has helped enhance the quality and clarity of the study.

Reviewer 2 Report
Comments and Suggestions for Authors
The aim of this retrospective study, conducted at a single tertiary center, was to refine prognostic stratification by identifying a very favorable subgroup within the favorable-risk group of patients with metastatic renal cell carcinoma (mRCC) who were treated with a tyrosine kinase inhibitor (TKI). The study analyzed 189 patients diagnosed with mRCC. Based on the selected criteria, 29 patients were categorized into the very favorable group, and 46 patients were categorized into the favorable subgroup. The authors found that patients in the very favorable group exhibited significantly longer median overall survival (OS) and progression-free survival (PFS) compared to those in the favorable subgroup. This finding highlights the heterogeneity within the IMDC favorable-risk category and underscores the need for further large-scale prospective studies.
Furthermore, in the era when immune checkpoint inhibitors (ICIs) are generally accepted as first-line therapy for mRCC, the authors suggest that there remains a subset of patients who benefit from TKIs as first-line therapy.
The manuscript includes two tables and two figures. It cites 20 references, which is relatively few but sufficient for a manuscript of this scope. Although the cohort of patients is small, the results are robust and could have a direct impact on the management of mRCC patients. This manuscript would be of interest to a broad audience of oncologists and urologists.
I recommend adding the number of patients at risk to the Kaplan-Meier curves.
Author Response

(The authors gave the same response as above.)

Reviewer 3 Report
Comments and Suggestions for Authors
The topic is very interesting, it is of great importance from the point of view of personalized therapy. That is why the manuscript involves important information.
I would like to suggest a few additions to the topic:
1. In the Introduction, I would suggest that little is written about the risk factors for kidney tumors. about TKI, combined with immuntherapy, and personlized therapy
2. What kind of risk factors were taken into account for the groups studied? Is there any information on familiar accumulation and genetic factors that could be related to the development of kidney tumors?
3. Reference should be added to the IMDC method and the Karnovsky assessment
4. Reference should be added regarding the TNM classification
5. Clinical patients are missing complete clinicopathological data:
tumor size, the type of surgery (partial or total nephrectomy)?
6. If there are any kind of associated tumors in metastatic cases were observed?
Author Response

(The authors gave the same response as above.)

Round 2
Reviewer 1 Report
Comments and Suggestions for Authors
concerns adequately addressed